# Inhaled corticosteroids do not adversely impact outcomes in COVID-19 positive patients with COPD: An analysis of Cleveland Clinic's COVID-19 registry

Payal Sen[1], Uddalak Majumdar[1], Joe Zein[1,2], Umur Hatipoğlu[1], Amy H. Attaway[1]*

1 Respiratory Institute, Cleveland Clinic, Cleveland, Ohio, United States of America, 2 Lerner Research Institute, Cleveland Clinic, Cleveland, Ohio, United States of America

* attawaa@ccf.org

## Abstract

Inhaled Corticosteroids (ICS) are commonly prescribed to patients with severe COPD and recurrent exacerbations. It is not known what impact ICS cause in terms of COVID-19 positivity or disease severity in COPD. This study examined 27,810 patients with COPD from the Cleveland Clinic COVID-19 registry between March 8th and September 16th, 2020. Electronic health records were used to determine diagnosis of COPD, ICS use, and clinical outcomes. Multivariate logistic regression was used to adjust for demographics, month of COVID-19 testing, and comorbidities known to be associated with increased risk for severe COVID-19 disease. Amongst the COPD patients who were tested for COVID-19, 44.1% of those taking an ICS-containing inhaler tested positive for COVID-19 versus 47.2% who tested negative for COVID-19 (p = 0.033). Of those who tested positive for COVID-19 (n = 1288), 371 (28.8%) required hospitalization. In-hospital outcomes were not significantly different when comparing ICS versus no ICS in terms of ICU admission (36.8% [74/201] vs 31.2% [53/170], p = 0.30), endotracheal intubation (21.9% [44/201] vs 16.5% [28/170], p = 0.24), or mortality (18.4% [37/201] vs 20.0% [34/170], p = 0.80). Multivariate logistic regression demonstrated no significant differences in hospitalization (adj OR 1.12, CI: 0.90–1.38), ICU admission (adj OR: 1.31, CI: 0.82–2.10), need for mechanical ventilation (adj OR 1.65, CI: 0.69–4.02), or mortality (OR: 0.80, CI: 0.43–1.49). In conclusion, ICS therapy did not increase COVID-19 related healthcare utilization or mortality outcome in patients with COPD followed at the Cleveland Clinic health system. These findings should encourage clinicians to continue ICS therapy for COPD patients during the COVID-19 pandemic.

## Introduction

The COVID-19 pandemic caused by infection with Severe Acute Respiratory Syndrome coronavirus 2 (SARS-CoV-2) is the greatest challenge the world has faced in the 21st century. A significant subset of infected patients are hospitalized due to severe pneumonia and may progress

**Data Availability Statement:** Data used for the generation of this research study includes human research participant data that are sensitive and

 

cannot be publicly shared due to legal and ethical restrictions by the Cleveland Clinic regulatory bodies including the Institutional Review Board and legal counsel. In particular, variables like the patient's address, date of testing, dates of hospitalization, date of ICU admission, and date of mortality are HIPAA protected health information and legally cannot be publicly shared. Since these variables were critical to the generation and performance of our statistical models, a partial dataset (everything except them) is not fruitful either because it will not help in efforts of academic advancement, such as model validation or application. We will make our data sets available upon request, under appropriate data use agreements with the specific parties interested in academic collaboration. Requests for data access can be made to mascar@ccf.org.

**Funding:** This study was funded by the National Institutes of Health – National Heart, Lung and Blood Institute Grant: K08 HL133381 (JZ).

**Competing interests:** The authors have declared that no competing interests exist.

**Abbreviations:** ACE2, Angiotensin Converting Enzyme 2; ACEI, Angiotensin-converting enzyme inhibitors; ARB, Angiotensin II Receptor Blockers; ARDS, acute respiratory distress syndrome; CCHS, Cleveland Clinic Health System; CI, confidence interval; COPD, chronic obstructive pulmonary disease; COVID-19, Corona Virus Disease 2019; ECMO, extra-corporeal membrane oxygenation; EHR, electronic health records; ICD-9, International Classification of Diseases Ninth Revision; ICS, inhaled corticosteroid; LABA, long-acting beta-agonist; LAMA, long-acting muscarinic antagonist; MERS, Middle East respiratory syndrome; OR, odds ratio; RSV, Respiratory Syncytial Virus; SARS-CoV-2, severe acute respiratory syndrome coronavirus 2; UK, United Kingdom.

to acute respiratory distress syndrome (ARDS) necessitating prolonged ICU stays, mechanical ventilation, or extra-corporeal membrane oxygenation (ECMO) [1, 2]. Unlike other comorbidities like diabetes, obesity and hypertension, the prevalence of chronic respiratory disease among patients with COVID-19 appears to be lower than the general population [3]. However, patients with underlying lung disease develop worse outcomes when infected with COVID-19, including increased rate of mortality [4–6]. It is hypothesized that these patients have less pulmonary reserve (due to reduced lung function, abnormal lung structure, and dysfunctional immunity) making them more susceptible to developing ARDS and poor clinical outcomes [7, 8].

The role of corticosteroids in SARS-CoV-2 infection has greatly evolved since the beginning of the pandemic. While systemic corticosteroids showed possible harm in previous coronavirus pandemics (SARS and Middle East respiratory syndrome [MERS]) [9, 10], a large randomized control trial known as RECOVERY demonstrated reduced mortality in hypoxemic COVID-19 patients treated with dexamethasone [11]. However, the role and benefit of ICS has been debated in the context of SARS-CoV-2 infection [12, 13]. ICS therapies in COPD reduce expression of the ACE2 (Angiotensin Converting Enzyme 2) receptor, which is highly expressed in the upper respiratory tract of humans as the point of entry for SARS-CoV-2 [14, 15]. Pre-treatment of human respiratory epithelial cells *in vitro* with budesonide, in combination with long-acting beta-agonist (LABA) and long-acting muscarinic antagonist (LAMA) bronchodilators, was shown to inhibit human coronavirus HCoV-229E replication and cytokine production [16]. Another study demonstrated that the inhaled corticosteroid ciclesonide blocks SARS-CoV-2 RNA replication *in vitro* and inhibits SARS-CoV-2 cytopathic activity, possibly reducing the risk and severity of SARS-CoV-2 [17]. ICS in combination with bronchodilators is recommended in patients with severe COPD who have frequent exacerbations or those with Asthma-COPD overlap syndrome [18]. However, ICS use must be weighed with the risk of side effects, including increased susceptibility to upper airway infections [19, 20], a higher prevalence of pneumonia, and alterations to the lung microbiome [21]. On the other hand, discontinuing ICS for fear of contracting COVID-19 could place patients at a higher risk for COPD exacerbations [22].

Currently there are very few studies analyzing the safety and efficacy of ICS in patients with COPD in regards to COVID-19 infection rate or disease severity [23–25]. We hypothesized that amongst patients with COPD who develop COVID-19, those who are on ICS therapy will have similar inpatient outcomes, mortality and healthcare utilization as those who are not on ICS.

## Methods

### Cleveland Clinic registry

Data on patients' demographics, medications, comorbidities, history of COVID-19 exposure, disease manifestation upon presentation, disposition, and outcomes were extracted from electronic health records (EHR) for all patients from the Cleveland Clinic COVID-19 registry [26]. Registry characterization and data collection reflect the clinical characteristics recently published on COVID-19 [27–31]. Uniform clinical templates were implemented across the Cleveland Clinic Health System (CCHS) using EHR to standardize the care of patients tested for COVID-19, and to facilitate data extraction. Data extraction from EHR (Epic®, Epic Systems Corporation, Wisconsin, USA) at the CCHS was performed manually by a trained research team and using predefined processes that have previously been published [32]. All data in the registry was fully anonymized and exempted from informed consent. This study and the registry were both approved by the Cleveland Clinic Institutional Review Board (IRB#20–391).

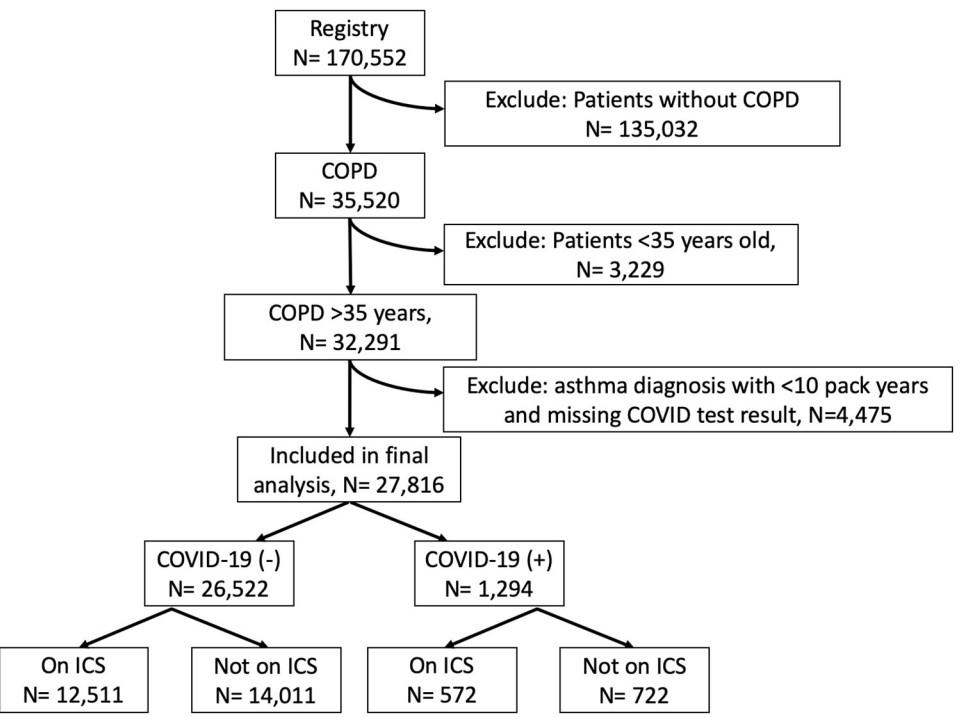

**Fig 1. Flowchart of patients included in our analysis.**

Testing for COVID-19 at Cleveland Clinic facilities is currently indicated for symptomatic patients (presence of fever, cough, shortness of breath or other symptoms) and those with chronic medical conditions. After March 21 2020, diarrhea was added to the qualifying symptoms [33].

## Subjects

Data on 170,552 individuals tested for COVID-19 at the Cleveland Clinic between March 8th, 2020 and September 16th, 2020 was available. Data was extracted from the EHR on September 20th, 2021. We limited our analysis to those with COPD who were 35 years and older because the diagnosis of COPD is unlikely in those less than 35 years old [34]. Additionally, to avoid having a biased sample, we excluded those with concurrent diagnosis of asthma and less than a 10 pack year smoking history. Out of the initial sample size of 170,552, 27,816 met our inclusion criteria (see Fig 1). Diagnosis of COPD was based on ICD-9 (491.x) and ICD10 codes (J41.0, J41.1, J41.8, J42, J43.1, J43.2, J43.8, J43.9, J44.0, J44.1, J44.9). Additional comorbidities (asthma, congestive heart failure, hypertension, diabetes mellitus) were also extracted by ICD10 codes and are listed in S1 Table.

## Laboratory confirmation

Nasopharyngeal and oropharyngeal swab specimens were collected and pooled for testing by trained medical personnel as previously described [33]. Infection with SARS-CoV-2 was confirmed by laboratory testing using the Centers for Disease Control and Prevention reverse transcription–polymerase chain reaction SARS-CoV-2 assay that was validated in the Cleveland Clinic Robert J. Tomsich Pathology and Laboratory Medicine Institute. This assay used an extraction kit (MagNA Pure; Roche) and 7500 DxReal-Time PCR System instruments

(Applied Biosystems) [35]. All testing was authorized by the Food and Drug Administration under an Emergency Use Authorization and in accordance with the guidelines established by the Centers for Disease Control and Prevention [33].

## Statistical analysis

COPD patients were analyzed for associations with COVID-19 positivity, and clinical outcomes in those that tested positive (including hospitalization, ICU admission, non-invasive ventilation and mechanical ventilation, and in-hospital mortality). Summary statistics included counts and percentages for categorical variables and means with standard deviations for continuous variables (which were all normally distributed). Data with missing dependent variables were excluded. Only 2 covariates had missing data which were smoking status (3.4%) and ethnicity (1.1%). Categorical variables were analyzed with chi-square tests and normally distributed continuous variables were compared using t-tests. Models were constructed choosing covariates known *a priori* to be associated with COPD and COVID-19 severity and identified from clinical experience and a review of the literature [36–42]. Binomial multivariate logistic regression was used to account for differences in clinical outcomes for COVID-19 infection, risk of hospitalization, risk of ICU admission, invasive mechanical ventilation, and in-hospital mortality, and included: Model 1 (adjusted for gender, race [African American, Caucasian, Hispanic, and other], and age), and Model 2 (adjustment for age, gender, race, smoking status [current versus non-current], comorbidities [asthma, diabetes mellitus, congestive heart failure, hypertension, obesity]). Our model compared current smokers to non-current smokers, the majority of which were former smokers, in order to save degrees of freedom in our model as never smokers represented less than 5% of the total cohort. Given the evolving nature of the COVID-19 pandemic and the development of new therapies [11], evidence to date has shown an improvement in mortality over time [43]. Therefore, our model also adjusted for the month of COVID-19 positivity. Additional studies were performed using the same models analyzing clinical characteristics and outcomes for COPD patients who required at least one course of oral corticosteroids (OCS) in the prior year (including prednisone, prednisolone, or methylprednisolone), the results of which are presented in S2–S4 Tables. Model fit was assessed using $R^2$ and C-index. All analyses were two-tailed, performed at a significance level of 0.05, and confidence intervals were 95%. R version 4.0.0 (The R Foundation for Statistical Computing, Vienna, Austria) were used for statistical analyses.

## Results

A total 27,810 patients diagnosed with COPD and tested for COVID-19 were included in the final analysis, of which 1,288 (4.6%) patients tested positive. Amongst the COVID-19 (+) cohort, 568 patients utilized ICS-containing inhalers and 720 did not utilize ICS. The demographic characteristics of the patients with COPD in the final analysis are summarized in Table 1. Among the COVID-19 (+) patients, 499 (38.8%) were males, and the mean age was 63.7±12.2 years. 676 (52.7%) of the COVID-19 (+) patients were characterized as obese. The COVID (+) cohort included 5.4% on ICS alone, 3.7% on LABA alone, 26.7% on LABA/ICS, 14.8% on LAMA, 4.9% on LAMA/LABA, and 12% on ICS/LAMA/LABA. 76.7% had been treated for an exacerbation with oral corticosteroids within the past year (inpatient or outpatient), which occurred prior to the hospitalization and unrelated to the diagnosis of COVID. Amongst the COPD patients who were tested for COVID-19, 44.1% of those taking an ICS-containing inhaler tested positive for COVID-19 versus 47.2% who tested negative for COVID-19 (p = 0.033).

**Table 1. Clinical characteristics of patients with COPD tested for COVID-19.**

| | COVID negative | COVID positive | p |
|---|---|---|---|
| N | 26522 | 1288 | |
| Demographics | | | |
| Male Sex (%) | 10947 (41.3) | 499 (38.8) | 0.079 |
| Race (%) | | | <0.001 |
| Black | 5074 (19.3) | 455 (35.8) | |
| Other | 526 (2.0) | 45 (3.5) | |
| White | 19689 (75.1) | 690 (54.3) | |
| Hispanic | 935 (3.6) | 81 (6.4) | |
| Age (in years) (mean (SD)) | 64.8 (13.9) | 63.7 (15.2) | 0.007 |
| BMI (mean (SD)) | 30.4 (8.2) | 32.0 (8.7) | <0.001 |
| Smoking status | | | <0.001 |
| Current | 4625 (18.0) | 115 (9.2) | |
| Former | 20917 (81.0) | 1126 (89.2) | |
| Never | 104 (0.4) | 5 (0.4) | |
| Comorbidities (%) | | | |
| Asthma | 6957 (26.2) | 379 (29.4) | 0.012 |
| Congestive heart failure | 6571 (24.8) | 328 (25.5) | 0.598 |
| Hypertension | 19997 (75.4) | 971 (75.4) | 1 |
| Diabetes | 14429 (54.4) | 723 (56.1) | 0.235 |
| Obesity | 11974 (45.3) | 676 (52.7) | <0.001 |
| Medications (%) | | | |
| Short acting beta agonist | 22460 (84.7) | 1104 (85.7) | 0.335 |
| ICS alone | 847 (3.4) | 69 (5.4) | <0.001 |
| LABA/ICS | 7555 (28.5) | 344 (26.7) | 0.18 |
| LAMA | 4871 (18.4) | 190 (14.8) | 0.001 |
| LAMA/LABA | 1584 (6.0) | 63 (4.9) | 0.122 |
| LABA | 1114 (4.2) | 48 (3.7) | 0.448 |
| ICS/LAMA/LABA | 4061 (15.3) | 155 (12.0) | 0.002 |
| **Inhaled corticosteroid containing inhaler** | 12511 (47.2) | 568 (44.1) | 0.033 |
| Oral corticosteroids | 20736 (78.2) | 988 (76.7) | 0.224 |
| Beta blocker | 17851 (67.3) | 809 (62.8) | 0.001 |
| ACE Inhibitor | 12422 (46.8) | 609 (47.3) | 0.776 |
| ARB | 7533 (28.4) | 387 (30.0) | 0.213 |

Data are presented as n (%) for categorical variables and mean [SD] for continuous variables. ICS = Inhaled corticosteroid, SABA = short acting beta agonist, SAMA = short acting muscarinic antagonist, LAMA = long acting muscarinic antagonist, LABA = long acting beta agonist. SAMA/SABA combination also included usage of nebulizer therapy. Inhaled corticosteroid containing inhaler represents any inhaler an ICS component, which include ICS, ICS/LABA, and ICS/LAMA/LABA combinations. Oral corticosteroids represent at least one course of steroids within the past year (prior to registry enrollment).

As shown in Table 2, COPD patients on an inhaler regimen that included ICS tended to be older (65.8±14.6 vs 62.1±15.5 years, p<0.001), and had higher rates of congestive heart failure (37.6% vs 16.1%, p<0.001), hypertension (82.7% vs 69.8%, p<0.001), and diabetes mellitus (61.9% vs 51.8%, p<0.001) compared to those not on ICS. Patients on ICS also had a higher prevalence of being on other inhalers such as SABA (short acting beta-2 agonists), LAMA, and

**Table 2. Clinical characteristics of all patients with COPD (inpatient and outpatient) who tested positive for COVID-19 based on ICS usage.**

|  | No ICS | ICS | p |
|---|---|---|---|
| N | 720 | 568 |  |
| **Demographics** | | | |
| Male gender (%) | 287 (39.9) | 212 (37.3) | 0.373 |
| Race (%) |  |  | 0.209 |
| Black | 257 (36.0) | 198 (35.5) |  |
| Other | 32 (4.5) | 13 (2.3) |  |
| White | 379 (53.2) | 311 (55.7) |  |
| Hispanic | 45 (6.3) | 36 (6.5) |  |
| Age (mean (SD)) | 62.0 (15.5) | 65.9 (14.7) | <0.001 |
| BMI (mean (SD)) | 31.8 (8.5) | 32.2 (9.0) | 0.401 |
| Smoking status |  |  | 0.092 |
| Current | 84 (11.7) | 56 (9.9) |  |
| Former | 620 (87.9) | 506 (90.7) |  |
| Never | 2 (0.3) | 3 (0.5) |  |
| Medications (%) | | | |
| LAMA | 24 (3.3) | 166 (29.2) | <0.001 |
| LAMA/LABA | 11 (1.5) | 52 (9.2) | <0.001 |
| Oral corticosteroids | 485 (67.4) | 503 (88.6) | <0.001 |
| **Comorbidities (%)** | | | |
| Asthma | 95 (13.2) | 284 (50.0) | <0.001 |
| Congestive heart failure | 115 (16.0) | 213 (37.5) | <0.001 |
| Hypertension | 502 (69.7) | 469 (82.6) | <0.001 |
| Diabetes | 372 (51.7) | 351 (61.8) | <0.001 |
| Obesity | 369 (51.7) | 307 (54.0) | 0.431 |
| **Outcomes (%)** | | | |
| Admission after positive | 42 (5.8) | 72 (12.7) | <0.001 |
| Month of COVID positivity (%) |  |  | 0.328 |
| March | 35 (4.9) | 30 (5.4) |  |
| April | 107 (14.9) | 84 (15.1) |  |
| May | 100 (13.9) | 102 (18.3) |  |
| June | 87 (12.1) | 64 (11.5) |  |
| July | 257 (35.7) | 180 (32.4) |  |
| August | 134 (18.6) | 95 (17.1) |  |

Data are presented as n (%) for categorical variables and mean [SD] for continuous variables. Month of COVID positivity represents the month during which the COVID test was positive. Oral corticosteroids represent at least one course of steroids within the past year (prior to registry enrollment).

LAMA/LABA. They were also more likely to be on beta blockers, ACEI and ARBs. Of those who tested positive, 375 patients (29%) required hospitalization.

For those hospitalized (see Table 3), 201 were on an inhaler regimen that included ICS and 170 were not on ICS. The BMI of COVID (+) patients on ICS was higher than those not on ICS (32.6±10.0 vs 30.5±8.7, p<0.001). When comparing the ICS users and ICS non-users, there was no significant difference in those who developed pulmonary embolism (12.9% vs 7.6%, p = 0.138), shock (15.9% vs 12.4%, p = 0.407), acute kidney injury (54.2% vs 45.3%, p = 0.107), acute liver failure (3.5% vs 5.3%, p = 0.549) or disseminated intravascular

**Table 3. Clinical characteristics and outcomes of hospitalized patients with COPD who tested positive for COVID-19 categorized by ICS usage.**

|  | **No ICS** | **ICS** | **p** |
|---|---|---|---|
| N | **170** | **201** |  |
| Demographics |  |  |  |
| Male sex (%) | 79 (46.7) | 77 (38.3) | 0.126 |
| Race (%) |  |  | 0.797 |
| Black | 72 (42.6) | 76 (38.4) |  |
| Other | 5 (3.0) | 7 (3.5) |  |
| White | 81 (47.9) | 104 (52.5) |  |
| Hispanic | 11 (6.5) | 11 (5.6) |  |
| Age (mean (SD)) | 67.1 (15.0) | 66.9 (14.0) | 0.921 |
| BMI (mean (SD)) | 30.5 (8.7) | 32.6 (10.0) | 0.034 |
| Smoking status |  |  | 0.074 |
| Current | 17 (10.2) | 14 (7.0) |  |
| Former | 146 (87.4) | 184 (92.5) |  |
| Never | 0 (0.0) | 1 (0.5) |  |
| Medications (%) |  |  |  |
| LAMA | 10 (5.9) | 76 (37.8) | <0.001) |
| LAMA/LABA | 6 (3.5) | 28 (13.9) | 0.001 |
| Oral corticosteroids | 124 (72.9) | 184 (91.5) | <0.001 |
| Comorbidities (%) |  |  |  |
| Asthma | 35 (20.6) | 93 (46.3) | <0.001 |
| Congestive heart failure | 47 (27.6) | 101 (50.2) | <0.001 |
| Hypertension | 144 (84.7) | 181 (90.0) | 0.162 |
| Diabetes | 112 (65.9) | 140 (69.7) | 0.507 |
| In-hospital conditions (%) |  |  |  |
| Pulmonary embolism | 13 (7.6) | 26 (12.9) | 0.138 |
| Sepsis | 45 (26.5) | 57 (28.4) | 0.773 |
| Pneumonia | 117 (68.8) | 157 (78.1) | 0.056 |
| Shock | 21 (12.4) | 32 (15.9) | 0.407 |
| Acute kidney injury | 77 (45.3) | 109 (54.2) | 0.107 |
| Acute liver failure | 9 (5.3) | 7 (3.5) | 0.549 |
| DIC and coagulopathy | 38 (22.4) | 40 (19.9) | 0.653 |
| In-hospital outcomes (%) |  |  |  |
| ICU admission | 53 (31.2) | 74 (36.8) | 0.303 |
| Endotracheal intubation | 28 (16.5) | 44 (21.9) | 0.237 |
| Mortality | 34 (20.0) | 37 (18.4) | 0.798 |
| Month of COVID positivity (%) |  |  | 0.863 |
| March | 7 (4.1) | 8 (4.2) |  |
| April | 17 (10.0) | 22 (11.5) |  |
| May | 27 (15.9) | 34 (17.8) |  |
| June | 31 (18.2) | 27 (14.1) |  |
| July | 49 (28.8) | 51 (26.7) |  |
| August | 39 (22.9) | 48 (25.1) |  |

Data are presented as n (%) for categorical variables and mean [SD] for continuous variables. DIC = disseminated intravascular coagulation. Month of COVID positivity represents the month during which the COVID test was positive requiring admission to the hospital. Oral corticosteroids represent at least one course of steroids within the past year (prior to registry enrollment).

**Table 4. Multivariate logistic regression analysis of COPD patients comparing those on ICS versus those not on ICS.**

| | COPD taking ICS versus COPD not taking ICS | | |
|---|---|---|---|
| | Unadjusted OR (95% CI) | Adjusted (model1) * OR (95% CI) | Adjusted (model 2) * OR (95% CI) |
| COVID positive | **0.89 (0.79–0.99)** | **0.85 (0.76–0.96)** | **0.85 (0.76–0.96)** |
| Hospital admission | **1.34 (1.09–1.65)** | **1.26 (1.02–1.55)** | 1.12 (0.90–1.38) |
| ICU admission[1] | 1.29 (0.84–1.99) | 1.38 (0.89–2.17) | 1.31 (0.82–2.10) |
| Ventilator[2] | 1.61 (0.79–3.32) | 1.37 (0.64–2.98) | 1.65 (0.69–4.02) |
| Mortality[1] | 0.90 (0.54–1.52) | 0.94 (0.54–1.64) | 0.80 (0.43–1.49) |

OR: Odds ratio, CI: Confidence interval, ICS: inhaled corticosteroid.

* Model 1 = Adjusted for gender, race, age.

* Model 2 = Adjusted for gender, race, age, smoking status (current versus former), comorbidities (asthma, obesity, diabetes mellitus, congestive heart failure, hypertension), and month of COVID positivity.

[1] Cohort includes only hospitalized patients.

[2] Cohort includes only ICU patients.

coagulation (20.2% vs 22.7%, p = 0.648). There was a trend towards increased ICD-10 diagnosis of pneumonia among ICS users (78.1% vs 68.8%, p = 0.056). Admission to the ICU (36.8% vs 31.2%, p = 0.303), rates of intubation (21.9% vs 16.5%, p = 0.237), and mortality (18.4% vs 20.0%) were not significantly different when comparing the groups.

As shown in Table 4, logistic regression analysis demonstrated that patients with COPD on ICS were less likely to test positive for COVID-19 compared to COPD patients not on ICS (unadj OR 0.89, CI 0.79–0.99), which held true when adjusted for gender, age, race (Model 1: OR 0.85, CI 0.76–0.96), and when additionally adjusted for comorbidities and month of COVID positivity (Model 2 OR: 0.85, CI 0.76–0.96). While COPD patients on ICS were more likely to be hospitalized (unadj OR 1.34, CI 1.09–1.65), which held true when adjusted for gender, race, and age (Model 1 OR: 1.26, CI 1.02–1.55), this association was not significant when adjusted for comorbidities and the month of diagnosis (Model 2 OR: 1.12, CI: 0.90–1.38). Clinical outcomes including ICU admission (Model 2 OR: 1.31, CI: 0.82–2.10), need for mechanical ventilation (Model 2 OR 1.65, CI: 0.69–4.02), and mortality (Model 2 OR: 0.80, CI: 0.43–1.49) were not significantly associated with ICS usage.

In order to further characterize our cohort based on COPD severity, additional analyses (S2–S4 Tables) were performed comparing COPD patients who had received at least one course of OCS in the prior year (prior to registry enrollment) to those who had not received OCS in the prior year. Clinical outcomes demonstrated an increased risk for hospital admission (unadj OR 1.70; CI: 1.26–2.33) and ICU admission (unadj OR 1.60; CI: 1.00–2.66) for those who had received OCS in the prior year. After model adjustment, hospital admission due to COVID-19 remained significantly associated with prior OCS usage (Model 2 OR 1.54; CI: 1.10–2.19).

## Discussion

In our large cohort study from the Cleveland Clinic healthcare system, we found that patients with COPD who were on ICS did not have worse outcomes from SARS-CoV-2 infection when compared to COPD patients not on ICS. ICS usage did not increase the risk for mortality, need for hospitalization, ICU admission or mechanical ventilation.

COPD is the fourth most common cause of death in the United States [44] and the third most common cause worldwide [45]. However, chronic respiratory diseases like COPD have not demonstrated increased prevalence for COVID-19 infection. Data from China and South

Korea illustrate that diabetes is far more prevalent than COPD among COVID-19 patients and is associated with worse outcomes [3, 46]. Our previous analysis of the Cleveland Clinic registry demonstrated that, while patients with COPD and COVID-19 had increased healthcare utilization, they did not have an increased risk for mortality compared to non COPD patients [47].

The reason why COPD is not prevalent amongst COVID-19 patients is unclear. The fact that telehealth studies have demonstrated feasibility to treat COPD patients with exacerbations as an outpatient may have led to greater adoption of telehealth in the management of COPD patients during the COVID-19 pandemic [48]. On the other hand, patients with COPD have less pulmonary reserve, reduced innate immunity to viral and bacterial infections, and also have a pro-thrombotic state with a number of associated cardiovascular comorbidities [7, 8]. Previous studies have shown that patients with COPD who develop respiratory viral illnesses due to influenza tend to have worse outcomes compared to those without COPD [49, 50]. Therefore, while the reason for reduced COVID-19 severity in COPD patients is unclear, the role of ICS as a potential therapeutic agent against COVID-19 requires further study [24].

While COVID-19 tends to disproportionately affect men more than women, our study comparatively had a higher number of female subjects. However, given that our study focused primarily on patients with COPD, this aligns with recent data reporting increased prevalence of COPD in women [51]. Women also have a higher symptom burden due to COPD among both smokers and non-smokers and are more likely to be hospitalized for COPD than males [52]. Therefore, it's possible that women were more likely to present to their doctors or be tested for COVID-19, which could have explained the higher proportion of females in our sample. We also note that patients who were taking ICS had more comorbidities, especially congestive heart failure, which raises several considerations. For one, severity of airflow obstruction is associated with higher rates of heart failure, and therefore usage of ICS may be a sign of increased COPD severity [53]. Second, a number of severe COPD patients have evidence for cor pulmonale or right ventricular failure. While the ICD9 and ICD10 codes we utilized for congestive heart failure did not include codes for right ventricular failure, it is possible that non-specific codes (i.e. I50.9, heart failure, unspecified) were utilized.

The benefits of ICS, particularly when used in combination with bronchodilators like LAMA and LABAs, include reductions in COPD exacerbations, improvement in COPD symptoms and better lung function [54]. Nonetheless, ICS may alter the lung microbiome, and is known to increase the risk of pneumonia in COPD patients [55]. Interestingly, our analysis showed that ICS use among patients with COPD was associated with an increased risk for pneumonia, however the finding was not statistically significant. Our registry also does not distinguish codes for viral pneumonia (which could be used to diagnose COVID-19 pneumonia) or bacterial pneumonia, and so it is unclear whether this was a consequence of COVID-19 or represented an additional superinfection. Because none of the other outcomes related to disease severity were worse in the ICS cohort in our study, we believe it is unlikely that this finding was an adverse consequence of ICS usage.

*In vitro*, ICS has been shown to attenuate the antiviral innate immune responses leading to delayed virus clearance [56, 57]. In previous novel coronavirus outbreaks (SARS, MERS), studies on ICS did not demonstrate benefit or harm [13], while systemic corticosteroids demonstrated possible harm [9, 10]. However, in patients diagnosed with COVID-19, dexamethasone significantly reduced 28-day mortality, particularly in those on supplemental oxygen or mechanical ventilation [11]. In the largest trial studying systemic corticosteroids (RECOVERY trial; n = 2104), 28-day mortality was 22.9% in the arm treated with dexamethasone compared to 25.7% in usual care (adj rate ratio 0.83, CI 0.75–0.93). Subgroup analysis demonstrated that mortality reduction was greatest in those requiring supplemental oxygen

(dexamethasone 23.3% vs. usual care 26.2%; rate ratio 0.82; CI 0.72–0.94) or mechanical venti-lation (dexamethasone 29.3% vs. usual care 41.4%; rate ratio 0.64, CI: 0.51–0.81).

Clinical data has previously demonstrated a protective effect of ICS in those patients who have frequent exacerbations due to COPD [18, 55]. Since about half of all COPD exacerbations are viral-induced [22], this suggests that ICS may attenuate the inflammatory viral response. This was previously demonstrated in studies of rhinovirus and RSV infections [56], as well as inhibition of viral replication and cytokine production for the coronaviruses responsible for the common cold [16, 58]. The inhaled corticosteroid ciclesonide also reduces replication and cytopathic effect of coronaviruses, including SARS-CoV-2, in cultured cells [59]. ICS has been shown to downregulate key virus-related genes in patients with asthma and COPD, including key SARS-CoV-2 genes. These include ACE2 and TMPRSS2 in asthma patients as well as ACE2 and ADAM17 in COPD patients [14, 15, 60].

The exact role of ICS in patients with COPD during the COVID-19 pandemic remains unclear. To the best of our knowledge, our study is the first to demonstrate both rates of COVID-19 positivity as well as inpatient outcomes comparing ICS users and non-users in terms of risk for hospitalization, ICU admission, need for mechanical ventilation or mortality. Schultze *et al.* reported increased risk of mortality risk among 148,557 patients with COPD on ICS but sensitivity analyses suggested this was from unmeasured confounding due to reduced baseline health status in patients on ICS [23]. Non-COVID-19-related deaths were also more common among COPD patients on ICS. Patients on triple therapy (LAMA, LABA and ICS combination) had higher mortality than patients on combined therapy (bronchodilator and ICS combination), even though the ICS exposure was the same. Bloom *et al.* studied inpatient clinical outcomes in patients with COPD and asthma who were hospitalized with COVID-19 in the UK. No benefit nor harm from ICS was demonstrated from their study of 12337 patients with COPD [61]. Aveyard *et* al. studied 8,256,161 patients with chronic lung disease from late January through April 2020, of whom 0.2% were hospitalized with COVID-19. They found that ICS was associated with a modest risk of severe COVID-19 independent of the underlying respiratory disease. The risk was reduced although not normalized when adjusted for comor-bidities and demographic factors [62]. Finally, a recent retrospective observational study from Colorado, USA, reported no effect of ICS on rates of testing positive for SARS-CoV-2 [25]. The authors did report a lower rate of SARS-CoV-2 positivity in patients on systemic cortico-steroid therapy. While our study demonstrated lower COVID-19 positivity in the population taking ICS, it may be due to heightened sensitivity to respiratory symptoms among COPD patients during the pandemic. Patients with COPD taking ICS are also likely to have a higher symptom burden at baseline. To answer the question whether ICS use confers reduced suscep-tibility to SARS-CoV-2 infection, additional prospective studies which include asymptomatic users of ICS at enrollment are needed.

Our additional analysis of outcomes related to OCS demonstrated that patients who had received OCS in the prior year were more likely to be admitted to the hospital for COVID-19. This has also been demonstrated in a recent meta-analysis of COPD patients with COVID-19 [63]. However, the fact that ICS did not impart an increased risk for healthcare utilization in our cohort of COPD patients highlights the safety of ICS in our population, and that more studies of ICS and its impact on COVID-19 disease severity are needed.

Our study has several limitations. This is a retrospective cohort study of a single center. Our registry is dependent on the local prevalence of SARS-CoV-2 which may be different com-pared to other regions of the world. Given the nature of the disease and patterns of behavior related to social distancing, the prevalence of the virus varies significantly based on geographic location. Our findings may also differ from other observational studies of COPD due to the fact that Cleveland Clinic and its regional facilities did not experience a surge that

overwhelmed hospitals during our study period [64]. In general, ICS is only recommended for the most severe COPD patients who are symptomatic and have frequent exacerbations, or those with asthma-COPD overlap syndrome [55]. Because our data is based on ICD-10 codes extracted from EHR, the cohort could not be sub-divided based on severity of COPD by GOLD stage. Similarly, patients who were on ICS could not be subdivided by the potency and dose of their ICS. Our study sought to reduce the effect of confounding from asthma by excluding those with a diagnosis of asthma and less than 10 pack year smoking history and those younger than 35 years old. While the overall number of patients in our study with comorbid asthma and COPD was 29.4% which is within the typical range based on population studies [65], the diagnosis of asthma was higher in the ICS cohort compared to those not on ICS [66], and therefore confounding from asthma could still be present. In general, while asthma patients tend to be younger and have less comorbidities than COPD patients [67], this was not the case in our study as our population taking ICS was older and had significantly more comorbidities. Therefore, based on our study design, we believe our population of patients on ICS truly represented those with COPD and not asthma alone. Finally, while our model included clinical characteristics like age, gender, race, smoking status and comorbidities, other prognostic factors not included such as laboratory or radiologic markers of end-organ damage may improve the prediction of outcomes due to COVID-19 [68], which is an area of ongoing research.

In conclusion, our study demonstrates that patients with COPD who are maintained on ICS and test positive for COVID-19 have similar outcomes to those who were not on ICS. This adds to the growing body of evidence that maintaining COPD patients on ICS is safe and should be continued during the COVID-19 pandemic.

## Supporting information

**S1 Table. ICD9 and 10 codes used for diagnosis of medical conditions or outcomes.**
(DOCX)

**S2 Table. Clinical characteristics of all patients with COPD (inpatient and outpatient) who tested positive for COVID-19 based on OCS usage.**
(DOCX)

**S3 Table. Clinical characteristics and outcomes of hospitalized patients with COPD who tested positive for COVID-19 categorized by OCS usage.**
(DOCX)

**S4 Table. Multivariate logistic regression analysis of COPD patients comparing those on OCS versus those not on OCS.**
(DOCX)

## Author Contributions

**Conceptualization:** Payal Sen, Amy H. Attaway.

**Data curation:** Amy H. Attaway.

**Formal analysis:** Amy H. Attaway.

**Investigation:** Payal Sen.

**Methodology:** Payal Sen, Joe Zein, Umur Hatipoğlu.

**Supervision:** Joe Zein, Umur Hatipoğlu.

**Writing – original draft:** Payal Sen, Uddalak Majumdar, Amy H. Attaway.

**Writing – review & editing:** Payal Sen, Uddalak Majumdar, Joe Zein, Umur Hatipoğlu, Amy H. Attaway.

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
