## [Decision Letter · Decision Letter 0]

6 Apr 2021

PONE-D-21-08569

Inhaled corticosteroids do not adversely impact outcomes in COVID-19 positive patients with COPD: An analysis of Cleveland Clinic’s COVID-19 Registry

PLOS ONE

Dear Dr. Attaway,

Thank you for submitting your manuscript to PLOS ONE. After careful consideration, we feel that it has merit but does not fully meet PLOS ONE’s publication criteria as it currently stands. Therefore, we invite you to submit a revised version of the manuscript that addresses the points raised during the review process.Both reviewers found your manuscript interesting. However, they raises some concenrs which are mainly attributed to methodological issues. Please ensure that your decision is justified on PLOS ONE’s publication criteria and not, for example, on novelty or perceived impact.

We look forward to receiving your revised manuscript.

Kind regards,

Stelios Loukides

Academic Editor

PLOS ONE

Journal Requirements:

2.  Thank you for providing the date(s) when patient medical information was initially recorded. Please also include the date(s) on which your research team accessed the databases/records to obtain the retrospective data used in your study."

3. Please provide a reference for your SARS-CoV-2 assay."

4. In your ethics statement in the Methods section and in the online submission form, please provide additional information about the data used in your retrospective study. Specifically, please ensure that you have discussed whether all data were fully anonymized before you accessed them and/or whether the IRB or ethics committee waived the requirement for informed consent. If patients provided informed written consent to have data from their medical records used in research, please include this information.

Reviewers' comments:

Reviewer's Responses to Questions

**Comments to the Author**

1. Is the manuscript technically sound, and do the data support the conclusions?

Reviewer #1: Yes

Reviewer #2: Partly

2. Has the statistical analysis been performed appropriately and rigorously? 

Reviewer #1: Yes

Reviewer #2: Yes

3. Have the authors made all data underlying the findings in their manuscript fully available?

Reviewer #1: Yes

Reviewer #2: Yes

4. Is the manuscript presented in an intelligible fashion and written in standard English?

Reviewer #1: Yes

Reviewer #2: Yes

5. Review Comments to the Author

Reviewer #1: The authors retrospectively analyzed the association between ICS use and COVID-19 related disease severity in patients with COPD. In COPD, the ICS is an important treatment option. This manuscript provides valuable information for continuing ICS treatment in COPD patients during the COVID-19 pandemic. However, due to its retrospective nature, this study may have several inherent limitations.

1. COPD includes heterogenous group of patients. It would be better to have information about the patient's lung function and exacerbation history.

2. The dose-dependent differences in ICS may be important in analyzing the outcomes.

3. In Table 1, why is the rate of ICS alone so high in COPD patients? It is possible that the study included patients with asthma alone.

4. There may be other factors associated with in-hospital outcomes of COVID-19. It would be better to describe possible prognostic factors not included in the analysis of this study as limitations.

5. The authors described that the diagnosis of COPD based on ICD codes. Please explain how other comorbidities were diagnosed.

Reviewer #2: This study adds data to several previous studies assessing the role of ICS during COVID-19 infection and disease. The study is well-written, the sample size adequate, the statistical analyses rigorous. The main issue is that this study doesn't add much to what is already known.

Major comments

- The same analyses done for ICS should be performed for OCS. To the best of my knowledge, no data so far has assessed ICS and OCS separately in COVID19 patients and assessed COVID-19 outcome based on ICS and OCS use.

- The majority of cohorts report a higher prevalence of male sex among people hospitalized with COVID19. How do the authors justify their finding of more females than males in their cohort?

- Were all the patients who were not current smokers, never smokers? I suspect some of them were former smokers. Please add this information and perform additional adjustments for never/former/current smoking status.

- Table 2: It is to be expected that patients on ICS, who were significantly older, have more comorbidities than patients not on ICS. What is the point the authors are trying to make? This does not add much information to what is already known.

- Table 3: The data on the higher prevalence of CHF in ICS users is interesting: please comment

- Table 4: Please add to the table the clinical characteristics of COPD patients taking and not taking ICS (age, sex, use of LABA/LAMA, use of OCS)

- The fact that the association between ICS in COPD and hospitalization was not maintained when adjusting for comorbidities indicates that the presence of comorbidities is the main determinant of the hospitalization, as already known. I am struggling to find the novelty of these findings. Please clarify.

- I assume most of the patients hospitalized for COVID-19 had pneumonia at admission. Please clarify the inclusion of pneumonia among the in-hospital outcomes.

Minor comments

- Lines 102-111: I suggest to mention these results briefly in the introduction and discuss them more extensively in the discussion section

- - “We hypothesized that ICS therapy in individuals with COPD is not associated with a higher risk of COVID-19 related 115 healthcare utilization or mortality.”. It would be better to start by saying what the authors hypothesize the ICS is going to do/be, instead of what they are not going to do/be.

6. PLOS authors have the option to publish the peer review history of their article (what does this mean?). If published, this will include your full peer review and any attached files.

Reviewer #1: No

Reviewer #2: **Yes: **Francesca Polverino, MD PhD

---

## [Author Response · Author response to Decision Letter 0]

9 May 2021

Response to the Reviewers:

We would like to thank the editor and the reviewers for their time and effort to review our work and provide constructive suggestions. In response, we have incorporated all of their suggestions in the revised manuscript. We believe that these modifications have significantly enhanced the quality of the submission. We provide a point by point response to each of the suggestions of the editor and the reviewers below. 

1. Comment. Please ensure that your manuscript meets PLOS ONE's style requirements, including those for file naming.

Response. Thank you for this suggestion. In response, we have adhered to all the formatting requirements described in https://journals.plos.org/plosone/s/file?id=wjVg/PLOSOne_formatting_sample_main_body.pdf and

2. Comment. Please also include the date(s) on which your research team accessed the databases/records to obtain the retrospective data used in your study.

Response. We again appreciate this suggestion and in response we have the date of access which was September 20th, 2021 (Page 5 Line 111). 

3. Comment. Please provide a reference for your SARS-CoV-2 assay.

Response. Thank you for this suggestion. We have provided the reference for the Emergency Use Authorization from the FDA authorizing the RT-PCR assay used in our study (Page 5 Line 125-126). The assay was developed by the Cleveland Clinic Robert J. Tomsich Pathology and Laboratory Medicine Institute and authorized by the Food and Drug Administration under an Emergency Use Authorization and in accordance with the guidelines established by the Centers for Disease Control and Prevention.

4. Comment. In your ethics statement in the Methods section and in the online submission form, please provide additional information about the data used in your retrospective study. Specifically, please ensure that you have discussed whether all data were fully anonymized before you accessed them and/or whether the IRB or ethics committee waived the requirement for informed consent. If patients provided informed written consent to have data from their medical records used in research, please include this information.

Response. Thank you for this comment. All patient data in the registry for our cohort was fully anonymized and exempted from informed consent. We have added this information to our Methods section (Page 4 Line 103-104).

5. Comment. We note that you have indicated that data from this study are available upon request. PLOS only allows data to be available upon request if there are legal or ethical restrictions on sharing data publicly. In your revised cover letter, please address the following prompts: a) If there are ethical or legal restrictions on sharing a de-identified data set, please explain them in detail. Please also provide contact information for a data access committee, ethics committee, or other institutional body to which data requests may be sent. b) If there are no restrictions, please upload the minimal anonymized data set necessary to replicate your study findings as either Supporting Information files or to a stable, public repository and provide us with the relevant URLs, DOIs, or accession numbers. We will update your Data Availability statement on your behalf to reflect the information you provide.

Response. We have discussed with our institution further, and our data availability statement with the necessary information is included below. 

Data Availability: Data used for the generation of this research study includes human research participant data that are sensitive and cannot be publicly shared due to legal and ethical restrictions by the Cleveland Clinic regulatory bodies including the Institutional Review Board and legal counsel. In particular, variables like the patient's address, date of testing, dates of hospitalization, date of ICU admission, and date of mortality are HIPAA protected health information and legally cannot be publicly shared. Since these variables were critical to the generation and performance of our statistical models, a partial dataset (everything except them) is not fruitful either because it will not help in efforts of academic advancement, such as model validation or application. We will make our data sets available upon request, under appropriate data use agreements with the specific parties interested in academic collaboration. Requests for data access can be made to mascar@ccf.org. 

Reviewers' Comments to Author:

We would like to thank the reviewers for their constructive suggestions based on a careful review of our work. We have carefully evaluated each of these suggestions and made modifications throughout the manuscript as suggested and provide a point by point response below. 

Reviewer 1

1. Comment. COPD includes heterogenous group of patients. It would be better to have information about the patient's lung function and exacerbation history.

Response. Thank you for this constructive feedback. Since the data for our study is based on ICD-10 codes extracted from EHR, detailed information on spirometry was not available and could not be added to our analysis. We recognize this as a limitation of our study and have included this in our Discussion (Page 18 Line 340-341). We have also included a list of our ICD-10 codes as a supplementary table (Supplementary Table 1). 

Our updated analysis now includes detailed outcomes related to the usage of oral corticosteroids in the year prior to COVID-19 infection which provides an additional surrogate for severity of COPD in our cohort of patients (Page 18 Line 326-331). 

2. Comment. 2. The dose-dependent differences in ICS may be important in analyzing the outcomes.

Response. Thank you for constructive feedback. As our study data utilizes ICD-10 codes documented in the EHR, we could not separate patients on the basis of dose/potency of ICS. This is an important limitation and an area of future study that we have acknowledged in our Discussion section (Page 18 Line 341-342). 

3. Comment. In Table 1, why is the rate of ICS alone so high in COPD patients? It is possible that the study included patients with asthma alone.

Response. We appreciate the reviewer’s careful review of our Table 1, and on further review we discovered there was an error in our dataset and that a substantial number of patients were documented as being on both ICS and ICS/LABA at the same time. This was a duplicate entry that we confirmed with coordinators of the registry, and these patients should have only been documented as being on ICS/LABA. This did not affect our analysis or the conclusions of our study because each of these patients was only counted once in the analysis, but it did impact the numbers in our Table #1 which are now corrected. Our revised Table 1 now demonstrates that ICS alone was used in 3.3% of patients (916/27810).

In terms of the possibility of patients having asthma alone, our analysis attempted to reduce this risk by excluding those with a diagnosis of asthma and less than 10 pack year smoking history. We also excluded all COPD patients younger than 35 years old as COPD is less likely to be a true diagnosis in patients younger than 35 years old. We have included these important aspects of our design in our Methods section (Page 5 Line 111-112) and Discussion section (Page 18 Line 342-344). 

4. Comment. There may be other factors associated with in-hospital outcomes of COVID-19. It would be better to describe possible prognostic factors not included in the analysis of this study as limitations.

Response. Thank you for this important feedback. While several prognostic factors of COPD have been studied, we are still understanding factors which affect clinical outcomes in patients hospitalized with COVID-19. These factors include clinical factors, such as age, demographics, hypoxia, as well as laboratory and radiologic markers (PMID 32845042). However, factors we adjusted for are nonetheless among the first ones which were well-known to significantly affect clinical outcomes in both COPD and COVID-19. Various risk prediction tools are being studied currently to phenotypically stratify and prognosticate COVID-19 and are an important area of future research, which we have noted in our Discussion (Page 19 Line 351-355).

5. Comment. The diagnosis of COPD based on ICD codes. Please explain how other comorbidities were diagnosed.

Response. Thank you for allowing us to clarify. Presence of comorbidities in the study patients was ascertained through billed ICD-10 codes as well. We have included a list of our ICD-10 codes as Supplementary Table 1. We have also mentioned this in the Methods section of our manuscript (Page 5 Line 116-118).

Reviewer 2

1. Comment. Major Suggestions: The same analyses done for ICS should be performed for OCS.

Response. We appreciate this suggestion and have now performed this analysis which is included in Supplementary Tables 2-4. The clinical outcomes of patients who had received at least one course of oral corticosteroids in the past year (not related to COVID-19) demonstrated increased risk for hospital admission (unadj OR 1.70; CI: 1.26-2.33) and ICU admission (unadj CI: 1.00-2.66). After model adjustment, hospital admission remained significant (adj OR 1.54; CI: 1.10-2.19). Given that treatment for COPD exacerbations with OCS is associated with disease severity in COPD, this suggests that exacerbations within the previous year could be associated with increased risk for healthcare utilization in severe COPD patients who develop COVID-19. 

2. Comment. Major suggestion: The majority of cohorts report a higher prevalence of male sex among people hospitalized with COVID19. How do the authors justify their finding of more females than males in their cohort?

Response: Thank you for allowing us to address this aspect of our study which was also an unexpected finding. We believe a potential explanation is the fact that overall, our cohort tended to have more females than males as demonstrated in our Table 1, where COPD males who were COVID negative represented 41.3% of the cohort and males who were COVID positive represented 38.8% of the cohort. Recently, COPD has become more prevalent in women (PMID 17673696) and also women are more likely to experience higher symptom burden and higher rates of hospitalization due to COPD (PMID 9150319). We believe these are two potential explanations for our findings and have included these considerations in our Discussion (Page 15 Line 257-260). 

3. Comment. Major suggestion: Were all the patients who were not current smokers, never smokers? I suspect some of them were former smokers. Please add this information and perform additional adjustments for never/former/current smoking status.

Response. Thank you for allowing us to clarify this. This information is available in our dataset which we have added to our Table 1. Former smokers represented the majority of the cohort (79.3%) compared to current smokers (17%), and never smokers or missing data represented 3.7%. To save degrees of freedom in our model, we did not adjust for never or missing smoking status alone as they represented less than 5% of the total cohort. We have updated our methods to state this (Page 144-146 Line 6) and updated our Table 1 with this additional information. 

4. Comment. Table 2: It is to be expected that patients on ICS, who were significantly older, have more comorbidities than patients not on ICS. What is the point the authors are trying to make? This does not add much information to what is already known.

Response. We appreciate the opportunity to clarify this aspect of our study. Our goal was to emphasize that patients who were on ICS represented a more severe cohort of COPD patients, who were also more likely to have comorbidities. We also note that since patients on ICS are older, they are at increased risk for worse clinical outcomes due to COVID-19. While the age difference is an important aspect, the mean difference between the two cohorts was 3.9 years which we do not believe would account for the significant increase in the number of comorbidities seen in the ICS cohort compared to the non ICS cohort. 

5. Comment. The main issue is that this study doesn't add much to what is already known.

Response. We appreciate this perspective from the reviewer and would like to take the opportunity to highlight the novelty of our study. We do believe that the role of ICS during the COVID-19 pandemic is still unclear and that further evidence is needed to demonstrate its safety and efficacy. While current studies are analyzing ICS usage as a treatment for COVID-19 with a possible signal for benefit (PMID: 33676591), it is not known if utilizing ICS in the early stages of the disease increases the risk for SARS-CoV-2 viral replication and shedding (PMID 15494274). To the best of our knowledge, our study is the only one which studied clinical inpatient outcomes, including risk of hospitalization, ICU admission, need for mechanical ventilation and mortality, in addition to the association of COVID-19 positivity. The fact that our study showed no evidence for harm in a well characterized cohort of COPD patients is reassuring that usage of ICS in the early stages of disease does not impart increased risk. 

We also feel that the data on the effect of ICS on COVID-19 outcomes are conflicting and therefore this remains an unsettled issue. Schultze et al (PMID: 32979987) reported increased risk of mortality among 148,557 patients with COPD on ICS but sensitivity analyses showed that this was from unmeasured confounding of the poorer baseline general health status of patients on ICS. Bloom et al (PMID: 33676593) studied inpatient clinical outcomes in patients with COPD and asthma who were hospitalized with COVID-19 in the UK. No benefit nor harm from ICS was demonstrated from their study with 12337 patients with COPD. Aveyard et al (PMID: 33812494) studied 8,256,161 English patients with chronic lung diseases from late January through April, of whom 0.2% were hospitalized with COVID-19. They found that ICS was associated with a modest risk of severe COVID-19 independent of the underlying respiratory disease. The risk was reduced though not normalized when adjusted for comorbidities and demographic factors. Our study therefore highlights important research gaps in the literature including the role of ICS in COVID-19 susceptibility in COPD patients, and risks and benefits of continuing ICS in COPD patients infected by SARS-CoV-2. 

Finally, our additional analysis of outcomes related to OCS, which was suggested by this reviewer, has revealed important clinical information. Given that patients who had received OCS in the prior year were more likely to be admitted for COVID-19 suggests that recent exacerbations of COPD increase the risk for COVID-19 healthcare utilization. This has also been demonstrated in a recent meta-analysis of COPD patients with COVID-19 (PMID: 32869011). However, the fact that ICS (also a marker of severe COPD) did not impart an increased risk for healthcare utilization in the same cohort of COPD patients highlights the safety of ICS in this population. 

6. Comment. Table 3: The data on the higher prevalence of CHF in ICS users is interesting: please comment. 

Response. Thank you for this observation. Because our study is based on ICD-10 codes, the increased prevalence of heart failure in patients on ICS raises several possibilities. For one, severity of airflow obstruction is associated with higher rates of heart failure (PMID 23727296). Second, a number of severe COPD patients have evidence for cor pulmonale or right ventricular failure. While our ICD9 and ICD10 coding for congestive heart failure did not include codes for right ventricular failure, it is possible that non-specific codes (i.e. I50.9, heart failure, unspecified) were utilized for these patients. We have included these possibilities in our discussion section (Page 15 Line 264-269). 

7. Comment. Table 4: Please add to the table the clinical characteristics of COPD patients taking and not taking ICS (age, sex, use of LABA/LAMA, use of OCS).

Response: We would like to clarify that the clinical characteristics of COPD patients taking and not taking ICS are included in Table 2 and Table 3 which include information on age, sex, and use of OCS. We have added information on LAMA and LABA/LAMA to Tables 2 and 3. 

8. Comment. The fact that the association between ICS in COPD and hospitalization was not maintained when adjusting for comorbidities indicates that the presence of comorbidities is the main determinant of the hospitalization, as already known. I am struggling to find the novelty of these findings. Please clarify.

Response. We believe that our analysis on OCS has clarified this aspect of our study, as receipt of OCS within the prior year was associated independently with increased healthcare utilization after adjustment for comorbidities in the same cohort of patients. If we consider ICS usage to be a marker of disease severity, there would also remain an independent association of ICS use and increased risk for healthcare utilization which we did not find in our study. 

9. Comment. Please clarify the inclusion of pneumonia among the in-hospital outcomes.

Response. Our goal was to provide information on clinically relevant conditions related to COVID-19, including sepsis, pulmonary embolism, acute kidney injury, acute liver failure, DIC and coagulopathy, shock, and outcomes including ICU admission, endotracheal intubation, and mortality. We have clarified the difference between conditions and outcomes in our updated Table 3. Given that the most common manifestation of COVID-19 is respiratory-related, we felt that an important clinical condition to include was diagnosis of pneumonia. This is especially important given ICS is associated with increased risk for pneumonia in COPD patients (PMID 22786484). However, we acknowledge that our registry does not distinguish billing codes for viral pneumonia (which could be used to diagnose COVID-19 pneumonia) or bacterial pneumonia, and therefore it is unclear whether this diagnosis was a consequence of COVID-19 or represented an additional superinfection. Because none of the other outcomes related to disease severity were worse in the ICS cohort, we believe it is unlikely that this finding was an adverse consequence of ICS usage. We have highlighted this aspect of our study in the discussion (Page 15-16 Line 275-277). 

10. Comment. Minor suggestion - Lines 102-111: I suggest to mention these results briefly in the introduction and discuss them more extensively in the discussion section

Response. Thank you for this suggestion. We have followed this suggestion and expanded our section on the RECOVERY trial in the Discussion section (Page 16 Line 287-291).

11. Comment. Minor suggestion – It would be better to start by saying what the authors hypothesize the ICS is going to do/be, instead of what they are not going to do/be.

Response. Thank you for this helpful suggestion. We have changed this sentence to “We hypothesized that amongst patients with COPD who also get COVID-19, those who are on ICS therapy will have similar inpatient outcomes, mortality and healthcare utilization as those who are not on ICS.”

---

## [Decision Letter · Decision Letter 1]

19 May 2021

Inhaled corticosteroids do not adversely impact outcomes in COVID-19 positive patients with COPD: An analysis of Cleveland Clinic’s COVID-19 Registry

PONE-D-21-08569R1

Dear Dr. Attaway,

We’re pleased to inform you that your manuscript has been judged scientifically suitable for publication and will be formally accepted for publication once it meets all outstanding technical requirements.

Kind regards,

Stelios Loukides

Academic Editor

PLOS ONE

Additional Editor Comments (optional):

Reviewers' comments:

Reviewer's Responses to Questions

**Comments to the Author**

1. If the authors have adequately addressed your comments raised in a previous round of review and you feel that this manuscript is now acceptable for publication, you may indicate that here to bypass the “Comments to the Author” section, enter your conflict of interest statement in the “Confidential to Editor” section, and submit your "Accept" recommendation.

Reviewer #1: All comments have been addressed

Reviewer #2: All comments have been addressed

2. Is the manuscript technically sound, and do the data support the conclusions?

Reviewer #1: Yes

Reviewer #2: Partly

3. Has the statistical analysis been performed appropriately and rigorously? 

Reviewer #1: Yes

Reviewer #2: Yes

4. Have the authors made all data underlying the findings in their manuscript fully available?

Reviewer #1: Yes

Reviewer #2: Yes

5. Is the manuscript presented in an intelligible fashion and written in standard English?

Reviewer #1: Yes

Reviewer #2: (No Response)

6. Review Comments to the Author

Reviewer #1: The authors have adequately addressed my comments and I feel that this manuscript is now acceptable for publication.

Reviewer #2: The authors have addressed most of my comments. Some comments could not be addressed mainly because the requested information was missing in the cohort.

7. PLOS authors have the option to publish the peer review history of their article (what does this mean?). If published, this will include your full peer review and any attached files.

Reviewer #1: No

Reviewer #2: **Yes: **Francesca Polverino

---

## [Editor Report · Acceptance letter]

24 May 2021

PONE-D-21-08569R1 

Inhaled corticosteroids do not adversely impact outcomes in COVID-19 positive patients with COPD: An analysis of Cleveland Clinic’s COVID-19 Registry 

Dear Dr. Attaway:

I'm pleased to inform you that your manuscript has been deemed suitable for publication in PLOS ONE. Congratulations! Your manuscript is now with our production department. 

Kind regards, 

on behalf of

Dr. Stelios Loukides 

Academic Editor

PLOS ONE